# Anionic Phospholipids Stimulate the Proton Pumping Activity of the Plant Plasma Membrane P-Type H^+^-ATPase

**DOI:** 10.3390/ijms241713106

**Published:** 2023-08-23

**Authors:** Laura C. Paweletz, Simon L. Holtbrügge, Malina Löb, Dario De Vecchis, Lars V. Schäfer, Thomas Günther Pomorski, Bo Højen Justesen

**Affiliations:** 1Department of Molecular Biochemistry, Faculty of Chemistry and Biochemistry, Ruhr University Bochum, 44780 Bochum, Germany; laura.paweletz@ruhr-uni-bochum.de (L.C.P.); malina.loeb@ruhr-uni-bochum.de (M.L.); thomas.guenther-pomorski@ruhr-uni-bochum.de (T.G.P.); 2Center for Theoretical Chemistry, Faculty of Chemistry and Biochemistry, Ruhr University Bochum, 44780 Bochum, Germany; simon.holtbruegge@ruhr-uni-bochum.de (S.L.H.); dario.devecchis@ruhr-uni-bochum.de (D.D.V.); 3Department of Plant and Environmental Sciences, University of Copenhagen, 1871 Frederiksberg C, Denmark

**Keywords:** H^+^-ATPase, lipid–protein interaction, liposome, molecular modeling, proton pump, reconstitution

## Abstract

The activity of membrane proteins depends strongly on the surrounding lipid environment. Here, we characterize the lipid stimulation of the plant plasma membrane H^+^-ATPase *Arabidopsis thaliana* H^+^-ATPase isoform 2 (AHA2) upon purification and reconstitution into liposomes of defined lipid compositions. We show that the proton pumping activity of AHA2 is stimulated by anionic phospholipids, especially by phosphatidylserine. This activation was independent of the cytoplasmic C-terminal regulatory domain of the pump. Molecular dynamics simulations revealed several preferential contact sites for anionic phospholipids in the transmembrane domain of AHA2. These contact sites are partially conserved in functionally different P-type ATPases from different organisms, suggesting a general regulation mechanism by the membrane lipid environment. Our findings highlight the fact that anionic lipids play an important role in the control of H^+^-ATPase activity.

## 1. Introduction

Membrane proteins of the P-type ATPase family are key players in the formation and maintenance of electrochemical gradients across cellular membranes through primary, active transport of ions. Consequently, they are involved in numerous vital biological processes including signal transduction, nutrient transport, and cell–cell communication [1,2,3]. Based on sequence similarity and substrate specificity, the P-type ATPase family is divided into five distinct subfamilies (P1–P5, for reviews see [4,5]). Prominent examples are the Na^+^/K^+^-ATPase (P2), which generates the electrochemical gradients for sodium and potassium, and the sarcoplasmic reticulum Ca^2+^-ATPase (P2), which pumps calcium from the cytosol into the lumen of the sarcoplasmic reticulum of skeletal muscle cells. Plasma membrane H^+^-ATPases belong to the P3 subfamily and are only found in plant, fungi, and protozoa (reviewed in [6,7]). They transport H^+^ ions out of the cell to generate and maintain an electrochemical H^+^ gradient across the plasma membrane, which drives nutrient uptake and is involved in intracellular pH regulation, stomatal opening, and cell growth [8].

In accordance with their vital cellular functions, P-type ATPases are commonly subject to tight regulation, which for members of the P3 subfamily is mediated through autoinhibitory regulatory domains (R-domains) located at the N- and C-termini [9,10]. The impact of these domains on pumping activity is controlled by both post-translational modifications and accessory proteins. Of the eleven H^+^-ATPases found in the model plant *A. thaliana*, the isoform 2 (AHA2) is the most extensively studied. Phosphorylation of residues in the C-terminal R-domain of AHA2 (e.g., S931, T947) can inhibit or induce binding of regulatory proteins that are believed to displace the autoinhibitory R-domains from the catalytic domain, resulting in activation of the pump [9,10,11,12,13]. In addition, lysophospholipids have been shown to activate the plant H^+^-ATPase by a mechanism involving both cytoplasmic terminal domains of the pump. More recently, the isolated C-terminal R-domain of AHA2 was shown to bind phosphatidylserine (PS) in a protein lipid overlay assay [14]. Further evidence for a role of phospholipids in the regulation of H^+^-ATPases comes from studies at the level of isolated membranes, detergent-extracted membranes, and partly purified proteins either in detergent micelles or as proteoliposome reconstituted microsomes [14,15,16,17,18]. In these studies, the presence of anionic lipids stimulated the activity of several plant and fungal plasma membrane H^+^-ATPases, but it remains unclear whether this mode of regulation involves the C-terminal regulatory domain.

In this study, we used a C-terminal truncated version of AHA2 that lacks the regulatory R-domain [9,10]. Removal of the R-domain renders the enzyme to be in a constitutively active state [19], enabling us to study the impact of the lipid environment on the activity of AHA2 through reconstitution of the purified protein into liposomes of defined phospholipid composition. Measurements on the H^+^-transport activity of the reconstituted AHA2 showed that it was sensitive to the lipid environment and specifically stimulated by PS and, to a lesser extent, by two other anionic phospholipids, phosphatidylglycerol (PG) and phosphatidic acid (PA). In concert with these observations, molecular dynamics (MD) simulations identified specific lipid interaction sites involving protein residues that are conserved across a wide range of P-type ATPases.

## 2. Results

### 2.1. Activity of C-Terminal Truncated AHA2 Is Stimulated by Anionic Phospholipids

To investigate the role of the lipid environment on the activity of AHA2 devoid of the R-domain, a C-terminal truncated version of the protein was purified and reconstituted into preformed liposomes composed of different binary lipid compositions (Figure 1B). Thin layer chromatography and SDS-PAGE analysis confirmed efficient removal of detergent as well as the lipid composition and protein content of each preparation (Appendix B, Figure A1). Estimation of membrane orientations of AHA2 in the different lipid compositions did not show any significant differences: about 80% or more of the proteins had the cytoplasmic portion orientated to the outside (Appendix B, Figure A2).

The resulting proteoliposomes were tested for proton pumping activity using ACMA as a fluorescent ΔpH probe (Figure 1A). For all proteoliposome preparations, immediate fluorescence quenching was observed after incubation with Mg^2+^-ATP, which was promptly recovered after the subsequent addition of the protonophore CCCP, demonstrating that the reconstituted AHA2 was functional and able to generate a pH gradient (exemplary trace shown in Figure 1C). The initial slope is used as an estimate for the proton pumping rate, as no significant transmembrane proton leakage is yet assumed due to buildup of the gradient over time. Analysis of the initial rates of proton pumping revealed pronounced differences in AHA2 activity between the different lipid compositions (Figure 1D). Whereas the addition of 30 mol% of the non-bilayer lipid phosphatidylethanolamine (PE) to phosphatidylcholine (PC) did not affect AHA2 activity relative to pure PC, a significant and specific high proton pumping rate was observed for AHA2 in the presence of anionic species at the same concentration (Figure 1D). Noticeably, an almost two times higher proton pumping rate was recorded for PS containing proteoliposomes compared to PG and PA (5.3-fold versus 2.2 to 2.4-fold compared to PC only). Similarly, AHA2 reconstituted in the ternary lipid mixture PC:PE:PS (45:45:10) displayed a 3.2-fold increase in the proton pumping rate (Figure 1D, Appendix A).

To further characterize the effect of anionic lipids on AHA2 activity, proteoliposomes with intermediate lipid compositions were prepared and tested for proton pumping activity using ACMA. As shown in Figure 2, an increasing amount of PA and PG in the proteoliposomes resulted in an increased AHA2 proton pumping activity relative to pure PC of 3.3 and 2.6-fold at 20 mol%, respectively. A further increase in the amount of PA and PG resulted in a slight decrease in AHA2 activity (albeit not statistically significant). In contrast, with increasing amounts of PS, the proton pumping rate of AHA2 increased 5.3-fold over PC alone, reaching maximal activity at ~30 mol% PS in the range tested. Collectively, these results indicate a specific stimulation of AHA2 proton pumping activity by anionic phospholipids (Appendix A). Furthermore, the stimulation of activity by PS shown for the truncated AHA2 is also found for the full-length AHA2 (Appendix C, Figure A3).

### 2.2. Anionic Phospholipids Interact Favorably at Distinct AHA2 Sites

To investigate the molecular underpinnings of the observed stimulation of the H^+^ pumping activity by PS, multi-microsecond MD simulations were carried out using the coarse-grained MARTINI model. AHA2 was simulated in two different lipid bilayers composed of PS:PC (10:90 molar ratio) and PS:PC:PE (10:45:45 molar ratio), respectively. The two-dimensional lipid densities around AHA2 (Figure 3A) revealed that, compared to the neutral lipids, anionic PS strongly accumulates in the annulus around the transmembrane domain of AHA2. The PS lipids preferentially accumulate at distinct sites (Figure 3A, sites A–E). The majority of the identified sites are located on the cytosolic side of the membrane (Figure 3A, sites A, B, D, and E).

Next, to more closely investigate the identified sites, the contacts between individual AHA2 residues and the phospholipid headgroups were analyzed (Figure 3B, Appendix A). All the lipid enrichment sites A–E were found in the contact analysis. Most of the identified PS-contacting residues are positively charged lysine or arginine residues. To provide even deeper insights into the lipid–protein interactions at the structural level, three-dimensional lipid densities were computed from the MD simulation trajectories (Figure 4). In congruence with the above results, well-defined density maxima of anionic PS lipids were consistently found at the distinct positions, with the most pronounced density at site A_1_. Notably, the 3D density analysis not only shows sharp density peaks for the PS headgroups but also partly for the associated lipid tails, which tend to become more disordered towards the membrane midplane. In the following, the 2D and 3D lipid densities are further analyzed and discussed in conjunction with the protein–lipid contacts to elaborate the positioning and importance of individual protein–lipid interaction sites in more detail.

The identified interaction site A_1_ is located close to the cytoplasmic end of transmembrane helix M10, just next to the (missing) R-domain, and site A_2_ is adjacent to it, located at the cytoplasmic end of M3 and M7. The two sites share the contact residues ^M7^K705, ^M7^L706, and ^M7^K707, whereas residues ^M10^R842 and ^M3^K238 are assigned to the individual sites A_1_ and A_2_, respectively. Site B is located at the unstructured loop connecting M1 and the actuator (A) domain, in proximity to a proposed solvent tunnel and the cytoplasmic proton entry site leading to the proton acceptor pair D684 and N106 [20,21,22,23]. The lipid contact analysis revealed ^M1^K57 and ^M1^K60 as contact residues for site B. Interestingly, both residues have previously been suggested as possible lipid interaction sites on the basis of MD simulations [20]. The identified exoplasmic interaction site C is close to the M3–M4 loop and encompasses lipid-contact residues ^M3^Q266, ^M3^R267, ^M3^R268, ^M4^Y270, and ^M4^R271. The nearby and conserved ^M1^D92/^M1^D95 were speculated by Pedersen et al., 2007 to be involved in proton release [21]. Since site C is in the exoplasmic leaflet, its relevance for AHA2 activation in asymmetric biological membranes is presumably low. Site D returned the single contact residue ^M6^K692. It is located near the cytoplasmic end of M6 and does not significantly favor PS over neutral lipids (Appendix A). Thus, protein–lipid interactions at site D are unlikely to be directly linked to stimulation of AHA2 by anionic lipids, but they might be of structural importance. Finally, interaction site E is located next to site D at the end of M6, with contact residue ^M6^R694. The positively charged arginine is surrounded by other charged residues (K692, D693, K696) and is located next to a water cavity proposed to be involved in cytoplasmic proton entry to the proton binding site (D684) [20,21,22,23]. The MD simulations yield similar results for interaction sites A_1_, A_2_, and E in both lipid mixtures studied (PS:PC and PS:PE:PC). For sites B, C, and D, the lipid densities and contact probabilities are lower in the ternary mixture, possibly implying that in a native-like lipid environment PS binding becomes more refined to sites A_1_/A_2_ and E. The most pronounced preferential lipid interactions are consistently found at site A_1_.

### 2.3. Binding Sites for Anionic Lipids Are Partially Conserved among Plant and Fungal H^+^-ATPases

To investigate whether the putative anionic lipid–protein contact sites A, B, C, D and E are conserved, protein sequences from a range of P-type ATPases were aligned (see Methods). For this analysis, sequences from two isoforms of *A. thaliana* proton pumps (AHA1-2), plant proton pumps from tobacco (*Nicotiana plumbaginifolia*) and rice (*Oryza sativa subsp. Japonica*), yeast proton pumps from Baker’s yeast (*Saccharomyces cerevisiae*) and *Neurospora crassa*, and rabbit Ca^2+^-ATPase (SERCA) and shark Na^+^/K^+^-ATPase (Na,K) were selected for sequence alignment.

In general, P-type ATPases overall show very low sequence similarity between structures (Figure 5 and Appendix A) [4,24]. From sequence alignment of the regions covering the lipid contact sites, the only interaction site fully conserved among all the analyzed structures is the ^M10^R842 in A_1_. Curiously, the site A_1_ has previously been proposed as a lipid interaction site in SERCA (R989) with a bound lipid (PE) identified in the crystal structure of 4UU1, 2AGV, 3AR3-7, and 3W5C [25,26,27,28]. Additionally, the R1003/R1011 in the pig/shark Na,K-ATPase (3WGV), located only 5 amino acids downstream from the fully conserved ^M10^R842 in AHA2, has also been identified as a sub site of a lipid binding site at M8–10 [29,30]. Both the individual interaction sites as well as several flanking residues were found to be well conserved among the sequences for the analyzed plant proton pumps; of the individual lipid interaction sites only the ^M3^K238 (A_2_) and the ^M7^K707 (A_1_/A_2_) were found to be different, with a threonine in rice PMA (A_2_) and an alanine in tobacco PMA (A_1_/A_2_). Still, a pair of lysines is located in M8/M9 that is supposed to be a lipid interaction site based on the crystal structure and native MS [31]. For the two yeast proton pumps, only the ^M1^K60 in site B was fully conserved, with several flanking residues found to be conserved in sites A_1_/A_2_, A_2_, B, and D/E. As expected, the number of conserved residues found in both the lipid–protein interaction sites, as well as adjacent residues, were considerably lower in the SERCA and Na,K pumps than among the analyzed proton pumps. For both pumps, the only fully conserved interaction site is ^M10^R842 at site A_1_, while site D/E is partially conserved at ^M6^R694.

## 3. Discussion

As integral membrane proteins, P-type ATPase interact intimately with membrane lipids, and several specific lipid interactions have been reported. For P-type ATPase cation pumps, one example of lipid interaction is the association of the *S. cerevisiae* Pma1 with lipid raft domains and its interaction with sphingolipids, required for delivery to the plasma membrane (for review see [33]). Particularly anionic phospholipids have been shown to affect the activity of several P-type ATPases, including the activity of Ca^2+^ and Na^+^/K^+^ transporting P2 ATPases as well as several H^+^ pumping P3 ATPases [16,34,35,36]. Previous work on the plant H^+^-ATPase AHA2 suggested a role for the R-domain in interacting with the lipid environment and in response controlling the activity of the pump [14,37,38]. We here show that a C-terminus truncated version of AHA2, devoid of the R-domain, still retains sensitivity towards the presence of anionic lipids. These results indicate that lipid interactions with the core of the protein directly regulate its activity.

Reconstitution of purified AHA2 into liposomes with defined lipid composition enabled the specific characterization of the potential role of lipids on the pump outside the complexity of the native environment. The proton pumping activity of the resulting AHA2-containing proteoliposomes was shown to vary depending on the presence of anionic lipids, with the strongest stimulation by PS. Within cells, PS is enriched along the secretory pathway, constituting 1.5 mol% of the glycerophospholipids in late secretory vesicles and 7 mol% in the plasma membrane, respectively [39]. Additionally, evidence has been presented that PS might localize to nanodomains within membranes [40]. Considering environmental conditions, the PS content of root cells is highly regulated in response to factors such as day–night cycles, drought–stress, and sugar–starvation [41,42,43]. Previous reports have been made on PS-induced stimulation of ATPase and/or proton pumping activity of detergent extracted membranes containing H^+^-ATPases from maize, *Zea mays*, rice, *Oryza sativa*, and mung bean, *Vigna radiata* [15,17,44], implying a general role of PS in regulating plant H^+^-pumps. Of the tested anionic lipids, PS showed the strongest stimulation of H^+^ pumping as compared to PA and PG, implying a specific lipid–protein interaction. Consequently, several regions with an increased anionic lipid density were identified using coarse-grain MD simulations of AHA2 embedded in different membrane compositions, each containing 10% of the anionic phospholipid PS in a background of neutral lipids. The identified sites were all located at the lipid–protein interface, and potential residues for anionic lipid–protein interaction sites were subsequently found by contact analysis. Of the five sites located at the cytoplasmic side, site B with ^M1^K57 and ^M1^K60 has previously been identified as possibly involved in the function and regulation of P-type ATPases [20]. A segment covering both residues is almost entirely conserved in the plant proton pumps as shown in the multiple sequence alignment (Figure 5C), with ^M1^K60 also being conserved in the two yeast proton pumps *S. cerevisiae* Pma1 and *N. crassa* Pma1. The two residues have previously been identified in the report on the refined crystal structure of AHA2 by Focht et al. (2017) as possible interaction sites for anionic phospholipids. Furthermore, MD simulations showed this region to partition into the membrane interface, inducing a local depression speculated to facilitate solvent access to the H^+^ binding site (D684-N106 pair) [20]. Similar observations have recently been made for the structures of *S. cerevisiae* Pma1 [45] and *N. crassa* Pma1, solved in the auto-inhibited E1 conformation [46]. In *N. crassa* Pma1, M1 is buried deeply within the membrane facilitating cytosolic access towards the H^+^ binding site, with K115, corresponding to ^M1^K60 in AHA2, oriented to the lipid headgroup–aqueous interface as a so-called snorkeling residue [46]. A similar proton access pathway is found in the structure of *S. cerevisiae* Pma1, where additionally the proton pumping cycle is found to involve movement of M1 towards the cytosolic leaflet of the bilayer in the E2P state [45]. Such re-arrangement of M1 is also supported by a proposed mechanism of proton transport in *N. crassa* Pma1 based on an SERCA homology model [46]. In this context, a possible role of the anionic lipid binding site B could be to aid in the movement of M1 towards the cytosolic leaflet, accelerating the conformational changes required for occlusion of the proton binding site followed by exposure to the exoplasmic site and thereby increasing the proton pumping rate.

In addition to the structural data detailed above, mutational studies on residues located in the region of site B have also been reported. An E167K mutation in the related *A. thaliana* Ca^2+^-ATPase ACA2, corresponding to ^M1^K60 in AHA2, resulted in a deregulated pump with activity similar to the activated state [47]. A similar result was observed for a P72A mutation in *N. plumbaginifolia* Pma2, corresponding to ^M1^P68 in AHA2, which is conserved in P3 ATPases and is responsible for the 90° kink in the M1 speculated to facilitate solvent access to the H^+^-binding site [41,48]. This mutation also removes the sensitivity and stimulation by lysophosphatidylcholine. Both residues have been proposed to be interacting with the autoinhibitory C-terminal regulatory domain, although while a similar P68S mutation in AHA1 resulted in increased proton pumping activity, it did not seem to inhibit interaction of the R-domain with the cytosolic domains [42,43]. Considering the MD simulations shown in this work, an alternative explanation for the observed activation could be an altered lipid interaction site in this region. This would also be in line with our observation of a lipid-induced proton pumping stimulation of AHA2 devoid of the R-domain. Further mutational studies are required to investigate such an effect.

Along with the above-mentioned residues at site B, mutations of the ^M4^D272A and ^M4^D275A located adjacent to the lipid–protein contact residues at site C seems to be the only examples of mutagenesis studies of the putative anionic interaction sites identified in this work [39]. However, the two mutations did not show any effect on the ability of AHA2 to complement the native yeast proton pump. Site C is the sole identified anionic lipid enrichment site in the exoplasmic leaflet and is located in close proximity to the proposed proton exit pathway with ^M4^D272 and ^M4^D275 at the periphery of the contact site buried deeper into the membrane near a putative cation binding site identified by [20,21,39]. Given the proximity of interaction site C to the putative proton exit pathway, one possible function of anionic lipids in this region could be to facilitate the attraction of protons during the exposure of the H^+^-binding site to the exoplasmic space. A similar role for the attraction of protons in the E1 conformation to the H^+^-binding site could also be envisioned for sites A_2_ and B.

The recently reported structures of the yeast proton pumps *S. cerevisiae* Pma1 and *N. crassa* Pma1 revealed a hexamer arrangement, confirming previous reports on the oligomerization of P3 type ATPases [40,45,46,49]. Similarly, several studies have also reported on the oligomerization of plant proton pumps [43,50]. Even though the exact role of oligomerization remains unknown, there are indications that for plant proton pumps, activation by 14-3-3 proteins involve the formation of hexamer complexes [43], while for the yeast proton pump *N. crassa* Pma1 autoinhibition is enhanced or possibly even dependent on the hexamer arrangement [46]. Based on the yeast proton pump structures, interaction sites A_1_, A_2_, and C could potentially be involved in mediating the formation of hexamers. In the structure of the *S. cerevisiae* Pma1, lipids in the exoplasmic leaflet were found to bind to M3/M7, at the interface between two monomer units, near the location of site C. Unfortunately, the lipid headgroups were not resolved, and therefore their exact composition remains unknown. MD simulations [46] on the *N. crassa* Pma1 identified two putative lipid binding sites near our site A_1_ (Site I) and site C (Site II) also located at the interface of two monomers. Whereas site I showed an accumulation of anionic PS, in accordance with the observations from the MD simulations on AHA2, site II showed preferred binding of PC. The identification of a putative PS binding site for *N. crassa* Pma1 fits well with reports showing the activity of the yeast proton pump to depend on the presence of anionic lipids [35] and the PS enrichment in *N. crassa* Pma1 containing polymer nanodiscs [51]. The effects of anionic lipids on the function and regulation of P3 ATPases could very well entail several of the above proposed interactions and might furthermore differ between yeast and plant proton pumps despite the similarities found in the putative anionic lipid binding sites. Based on the putative lipid binding sites identified in this work, future mutagenesis studies of AHA2 could help to further elucidate the mechanisms involved in the PS mediated stimulation of proton pumping of plant plasma membrane P3 ATPases.

## 4. Materials and Methods

### 4.1. Materials

Phospholipids 1-palmitoyl-2-oleoyl-*sn*-glycero-3-phosphatidylcholine (PC), 1-palmitoyl-2-oleoyl-*sn*-glycero-3-phosphatidylglycerol (PG), 1-palmitoyl-2-oleoyl-*sn*-glycero-3-phosphatidylethanolamine (PE), 1-palmitoyl-2-oleoyl-*sn*-glycero-3-phosphatidylserine (PS), 1-palmitoyl-2-oleoyl-*sn*-glycero-3-phosphatidic acid (PA), and 1,2-dipalmitoyl-*sn*-glycero-3-phosphoethanolamine-N-(7-nitro-2-1,3-benzoxadiazol-4-yl) (N-NBD-PE) were purchased from Avanti Polar Lipids Inc. (Birmingham, AL, USA). N-dodecyl-β-maltoside (DDM) and n-octyl-β-d-glucoside (OG) were obtained from Glycon (Luckenwalde, Germany). The ionophores valinomycin and CCCP, the pH-sensitive dye ACMA, and all other chemicals and regents were from Sigma-Aldrich (München, Germany), if not stated otherwise. ACMA was dissolved in dimethylsulfoxide; valinomycin in ethanol.

### 4.2. Preparation of Arabidopsis Thaliana H^+^-ATPase Isoform 2

A 73 amino acid C-terminal truncated version of *A. thaliana* H^+^-ATPase isoform 2 (designated AHA2), containing a hexahistidine (6 × His) and a SNAP^®^ tag at the N-terminal end of the protein, was overexpressed in the *Saccharomyces cerevisiae* strain RS-72 (*MATα*, *ade*1-100, *his*4-519, *leu*2-3,112) and purified according to previously published protocols [19,52]. All buffers contained 0.2 mM phenylmethylsulfonyl fluoride and 2 µg mL^−1^ pepstatin. The cells were lysed by mixing with glass beads, and the protein was solubilized and purified with DDM at a detergent:protein (*w*/*w*) ratio of 1:3 using batch-binding to a Ni^2+^-NTA resin. The purified protein was finally concentrated to 1–10 mg mL^−1^ using centrifugal concentrators with a cut-off at 100 kDa (Vivaspin 100, GE Healthcare, Chicago, IL, USA), frozen in liquid nitrogen and stored at −80 °C in storage buffer containing 50 mM Mes-KOH (pH 7), 50 mM KCl, 20% (*v*/*v*) glycerol, 1 mM EDTA and 1 mM DTT supplemented with 0.04% (*w/v*) DDM until further use.

### 4.3. Liposome Preparation

Liposomes were prepared by re-hydration of a thin lipid film, followed by freeze-thawing and manual extrusion. Briefly, binary lipid mixtures were prepared by mixing appropriate volumes of the lipid stock solutions in chloroform/methanol (2/1, *v*/*v*) and trace amounts (0.5 mol%) of fluorescent marker lipid N-NBD-PE in a glass tube. In addition to binary lipid mixtures, a ternary lipid mixture containing 45:45:10 PC:PE:PS was included in the set-up. The solvent was removed using a rotary evaporator (30 min at 200 mbar; 30 min at 100 mbar; 0 mbar for at least 2 h). The lipid film (10 mg) was re-hydrated in 667 µL reconstitution buffer (20 mM MOPS-KOH, pH 7, 50 mM K_2_SO_4_) by vortexing in the presence of a glass bead (5 mm diameter) for 5 min above phase transition temperature of the lipids (PE 30 °C; for all other room temperature), yielding a final lipid concentration of 15–20 mM. The vesicle suspension was further processed by five freeze-thawing cycles (90 s in liquid nitrogen followed by 90 s in a 60 °C water bath) and extrusion (21 times) through a stack of polycarbonate membranes (pore size 200 nm) using a mini-extruder (Avanti Polar Lipids). The vesicles made from PC:PE were unstable (lipids felt out of solution after short term storage), possibly because of an increased amount of PE in the membrane and the increased phase transition temperature. Thus, in this case, all steps (hydration and extrusion) were performed above 30 °C without storage at 4 °C. Liposome solubilization by OG was monitored by measuring light scattering of the liposome-containing solution at 600 nm using a fluorometer (PTI-Quantamaster 800, Horiba, Benzheim, Germany), thereby determining the ‘onset’ and ‘total’ solubilization conditions [53]. Detergent was added stepwise (2 µL of 250 mM OG in reconstitution buffer) to the liposome solution and the sample was stirred for 1 min before measuring the scattering. In order to calculate the concentration of OG required to reach the halfway point between saturation and complete solubilization of the vesicles, the scattering data as a function of OG concentration were fitted with a Boltzmann equation (Equation (1)) using a Python script,
(1)$$y=Bottom+\frac{Top-Bottom}{1+e^{\frac{x-V50}{slope}}}$$
where *Top* is the starting plateau of solubilization curve (maximal scattering of intact vesicles), *Bottom* is the final plateau of solubilization curve (minimal scattering of detergent-lipid micelles), *V*50 is the point of inflection, and slope refers to the slope at the point of inflection.

### 4.4. Vesicle Reconstitution

To facilitate the insertion of AHA2 into liposomes, preformed liposomes (4–5 mM) were solubilized by an amount of OG that was just sufficient to result in a half-maximal scattering change at 600 nm (‘Turning Point’). Purified AHA2 (12.5 μg) was added at a protein-to-lipid ratio of 1:60 (*w*/*w*). The protein/lipid/detergent mixture was subjected to gel filtration (Sephadex G-50 Fine, 3 mL packed in 2 mL disposable syringes) by centrifugation (180× *g*, 8 min). The eluate was incubated for 60 min at room temperature with 50 mg of prewashed SM-2 Bio-Beads (Bio-Rad Laboratories, Hercules, CA, USA) under end-over-end rotation to ensure detergent removal.

### 4.5. ATP-Dependent Proton Transport Assay

H^+^ pumping by AHA2 into the vesicles was measured as the initial rate of ACMA fluorescence quenching [54]. Proteoliposomes (50–200 µM) were added to 987–957 µL ACMA buffer (20 mM MOPS-KOH, pH 7.0, 50 mM K_2_SO_4_, 3 mM ATP, 1 µM ACMA, and 62.5 nM valinomycin). H^+^ pumping was initiated by the addition of MgSO_4_ (3 mM final concentration), and the H^+^-gradient dissipated by the addition of 5 μM CCCP. Fluorescence quenching was recorded over a period of 600–1200 s at 480 nm (excitation 412 nm, slit width 2 nm, resolution, 0.1 s) at 23 °C using a PTI-Quantamaster 800. Fluorescence traces were normalized to the intensity measured directly after addition of MgSO_4_. Recorded traces were analyzed with a customized python script with the slope of a linear curve fitted to the first 30 s after magnesium addition used as a measure for proton pumping activity. Traces that upon magnesium addition did not show a drop in fluorescence (signal before CCCP addition was less than 10% below the signal after CCCP addition) were considered inactive and excluded from analysis.

### 4.6. Protein Orientation Assay

AHA2 orientation in liposomes was determined by site-specific labeling of the SNAP-tag. Proteoliposomes (10 µL, ~500 ng AHA2) were incubated for 2 h either sequentially, first with 20 pmol membrane impermeable SNAP dye (10 µM SNAP-Surface^®^ 488, New England BioLabs Inc., Ipswich, MA, USA) followed by 20 pmol membrane permeable SNAP dye (10 µM SNAP-Cell^®^ 647-SiR, New England BioLabs Inc.) in reconstitution buffer supplemented with 1 mM DTT or with each dye separately. Samples were analyzed by SDS-PAGE using 12% gels and visualized on a ChemiDoc XRS Imaging System (Bio-Rad Laboratories GmbH, München, Germany) using the Image Lab™ software (https://www.bio-rad.com/zh-cn/product/image-lab-software?ID=KRE6P5E8Z&WT_mc_id=211202033049&WT_srch=1&WT_knsh_id=cr180107&gclid=EAIaIQobChMI0eKynPLxgAMVlZhmAh2HUArSEAAYASABEgJPdvD_BwE) and pre-programmed option for Coomassie stained gels or Alexa488/Alexa647 fluorophores, respectively [55].

### 4.7. Other Analytical Techniques

Phospholipid phosphorus was assayed after heat destruction in presence of perchloric acid by the method of Bartlett (1959) [56]. To check detergent removal and lipid composition, vesicles were analyzed by thin-layer chromatography using chloroform: methanol: ammonium hydroxide (63:35:5, *v*/*v*/*v*). Detergent and lipid standards were chromatographed on the same plate and applied without prior extraction by chloroform/methanol. For visualization, plates were stained with primuline (0.005% in acetone: water, 8:2, *v*/*v*) and photographed under long-wave UV light (Biorad ChemiDoc XRS Imaging System).

### 4.8. Data Analyses

To analyze the kinetic data, customized algorithms were developed using python version 3.8.8 [57]. Data are presented in form of boxplots, with median and 25/75% quantiles of experiments performed in at least duplicates, but for statistical analysis the mean ± S.E is used and given in sup Appendix A. One-way analysis of variance was performed using Tukey’s honestly significantly differenced (HSD) test employing the python packages scipy.stats [58], statsmodels [59], and bioinfokit [60]. The *p* values < 0.05 were interpreted as statistically significant. Based on the results, significance was indicated as following; with no statistical significance (ns) or statistically significant; *p* > 0.05, * *p* < 0.05, ** *p* < 0.01, *** *p* < 0.001.

### 4.9. Molecular Dynamics (MD) Simulations

For all computational studies the coarse-grain MARTINI 2.2 [61,62,63,64] force field was employed. The atomistic structure of AHA2 was retrieved from the protein data bank (PDB) as chain B of entry 5KSD [20], lacking the C-terminal autoregulatory R-domain. The structure was coarse-grained with *martinize* 2.6 and an elastic network was applied to stabilize the backbone conformation [65] by connecting backbone beads within a cut-off of 9 Å with harmonic potentials with a force constant of 500 kJ/mol/nm^2^. The protein was embedded separately in two lipid bilayers with *insane* [66], containing mixtures of PS:PC (10:90 molar ratio) or PS:PE:PC (10:45:45 molar ratio). Each bilayer contained 524 lipids in total, distributed symmetrically in both leaflets. All simulations were performed with GROMACS 2020.1 [67,68,69] at constant temperature (310 K) and pressure (1 bar) using periodic boundary conditions. The two systems were solvated and neutralized with 150 mM NaCl. Each system was energy minimized using steepest-descent and equilibrated with position restraints on all protein beads with a force constant of 1000 kJ/mol/nm^2^. Three 3 µs production runs were started from the equilibrated systems using different random seeds for generating the initial velocities of the particles at 310 K.

The “New-RF” MD parameters recommended by de Jong et al. [70] were used. A leap-frog algorithm was employed for integrating the equations of motion with a time step of 20 fs. Lennard-Jones 6–12 interactions were cut-off after 1.1 nm and shifted to be zero at that interparticle distance. Long-range Coulomb interactions beyond 1.1 nm were treated using a reaction field with the dielectric constant set to infinity. Velocity rescale thermostats with a coupling time constant of 1 ps were applied separately for the protein, the lipid membrane, and the solvent to maintain constant temperature of 310 K. Constant 1 bar pressure was maintained with a Parrinello–Rahman barostat with a coupling constant of 12 ps and compressibility of 0.0003 bar^−1^, separately for the membrane plane (xy-plane) and its perpendicular axis (z-axis).

### 4.10. MD Analyses

Overall translation and rotation of the protein were removed prior to analyses by structural alignment to the starting structure of the simulations. Analyses were conducted for each trajectory separately, and for the concatenated trajectories of the three analogous replicas. Contacts between protein residues and lipid headgroups were analyzed with GROMACS 2020.1 and custom Python scripts. A contact was counted when at least two beads, one from the protein residue and one from a lipid headgroup, were within 0.55 nm from one another and no other lipid was closer to the protein residue. The contacts were normalized by the number of timeframes to obtain the probability of finding a protein–lipid contact at any time. Two-dimensional lipid headgroup densities were computed with the GROMACS tool gmx densmap (grid spacing: 0.02 nm), separately for each leaflet and lipid species. Lastly, three-dimensional lipid densities were computed with MD Analysis [71,72] considering all PS beads or only PS headgroup beads (grid spacing: 0.1 nm).

### 4.11. Multiple Sequence Alignment

Protein sequences were downloaded from the UniProt knowledgebase database (access January 2021) and aligned with the tool Clustal Omega [73,74]. Accession numbers are as follows: P20649 and P19456, for the H^+^-ATPase isoforms 1 and 2, respectively, from *Arabidopsis thaliana*. Q42932, for the H^+^-ATPase isoform 2 from *Nicotiana plumbaginifolia*. Q7XPY2, for the H^+^-ATPase from *Oryza sativa subsp. Japonica*. P05030, for the H^+^-ATPase isoform 1 from *Saccharomyces cerevisiae*. P07038, for the H^+^-ATPase from *Neurospora crassa*. P04191, for the SERCA pump from *Oryctolagus cuniculus* and Q4H132, for the Na,K-ATPase from *Squalus acanthias*.

## Figures and Tables

**Figure 1 ijms-24-13106-f001:**
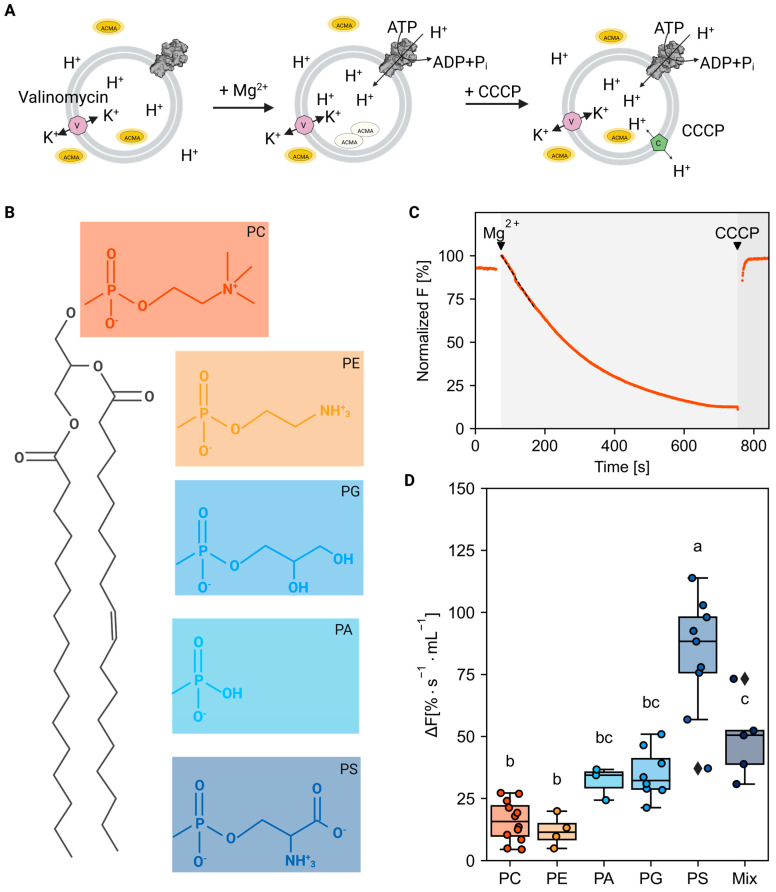
Effect of lipid bilayer composition on AHA2 proton pumping activity. (**A**) Illustration of the proton pumping assay on proteoliposomes with reconstituted AHA2. The accumulation of protons inside the vesicles was determined by measuring the fluorescence quenching of 9-amino-6-chloro-2-methoxyacridine (ACMA) as a fluorescent ∆pH probe. Reactions were started by the addition of Mg^2+^ to ATP-containing buffer. After reaching saturation conditions, the H^+^ gradient was disrupted by the addition of the protonophore m-chlorophenylhydrazon (CCCP). Valinomycin was always present to mediate K^+^ exchange and prevent the build-up of a transmembrane electrical potential. (**B**) Lewis structures of the lipids used. (**C**) Representative ACMA fluorescence trace on proteoliposomes with reconstituted H^+^-ATPase in PC liposomes. The initial slope (dashed line) was taken as a measure for the proton pumping rate. (**D**) Dot plot showing the initial rates of proton pumping of AHA2 reconstituted in liposomes composed of pure PC and PC in mixture with the indicated phospholipids (30 mol%); the mix contained PC:PE:PS (45:45:10). Colors are the same as in B, except for the mix. Letters above box plots indicate significant differences determined by Tukey’s HSD test (*p* < 0.05). Box plot center lines show the medians. Box limits indicate the 25th and 75th percentiles. Whiskers are extended to the highest and the lowest values. Data are based on at least three independent reconstitutions measured three times (see number of reconstitutions in Appendix A).

**Figure 2 ijms-24-13106-f002:**
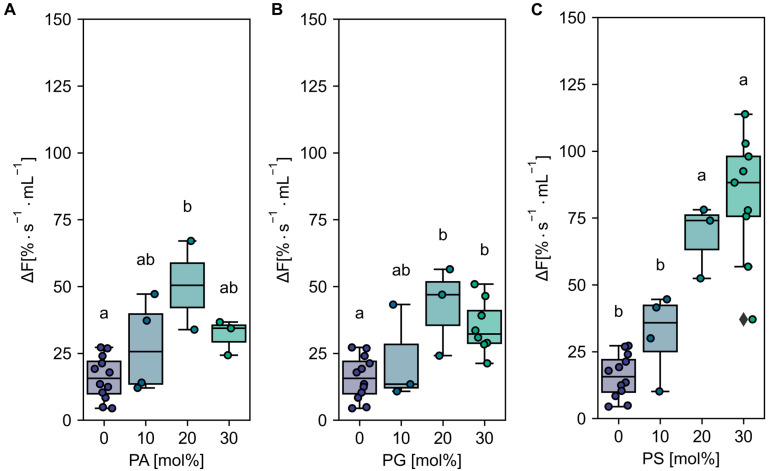
Effect of anionic lipids on AHA2 activity in proteoliposomes. AHA2 was reconstituted into liposomes composed of PC in mixture with the indicated amounts of anionic lipids, namely PA (**A**), PG (**B**), and PS (**C**). Dot plots show the initial rates of proton pumping; 0 and 30 mol% are listed for comparison and are the same data sets as in Figure 1. Letters above box plots indicate significant differences determined by Tukey’s HSD test (*p* < 0.05). Box plot center lines show the medians. Box limits indicate the 25th and 75th percentiles. Whiskers are extended to the highest and the lowest values. Data are based on at least two independent reconstitutions measured two to four times (see number of reconstitutions in Appendix A).

**Figure 3 ijms-24-13106-f003:**
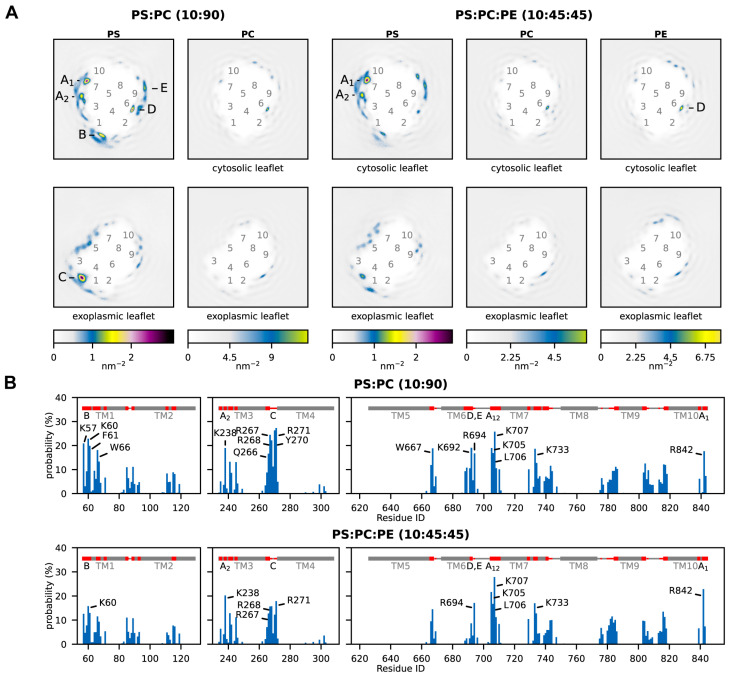
Phospholipid–AHA2 interactions from MD simulations. (**A**) Two-dimensional density of lipid headgroups, plotted separately for each leaflet. The color scales (bottom) are adjusted such that they reflect the molar ratio of the different lipid species in the bilayer. Labels 1–10 mark the approximate locations of the transmembrane helices; labels A–E mark local density maxima. (**B**) Contact probabilities of individual AHA2 residues with PS headgroups. Residues with contact probabilities above 15% are labeled. The secondary structure of AHA2 is depicted in grey at the top, with residues with ≥5% contact probability colored red and corresponding density maxima labeled A–E.

**Figure 4 ijms-24-13106-f004:**
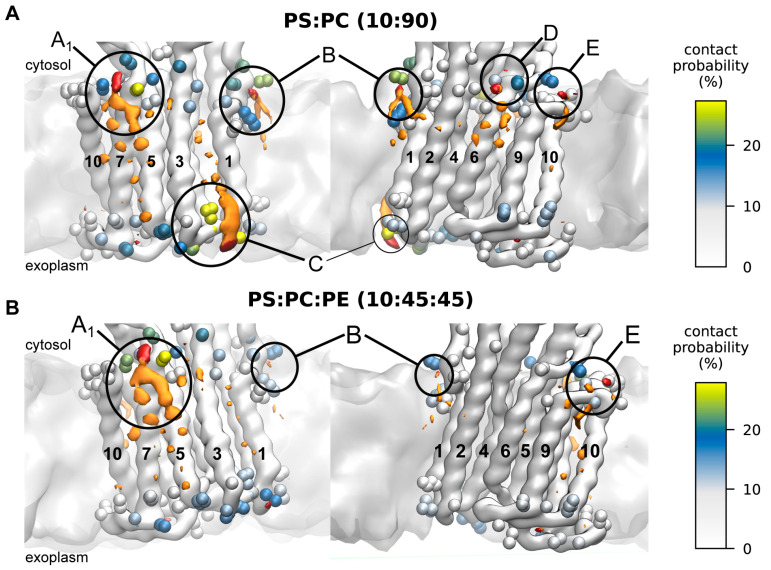
Phospholipid–AHA2 interaction sites within the membrane plane. Protein–lipid interaction sites shown from two perspectives, with isosurfaces of the three-dimensional density of PS headgroup beads shown in red (isovalue 3 nm^−3^) and the rest of PS in orange (isovalue 4 nm^−3^) for the two compositions tested, PS:PC (**A**) and PS:PC:PE (**B**), respectively. AHA2 residues with ≥5% lipid contacts are shown as beads, colored by contact probability (color bar at the right). The density maxima A–E are labeled.

**Figure 5 ijms-24-13106-f005:**
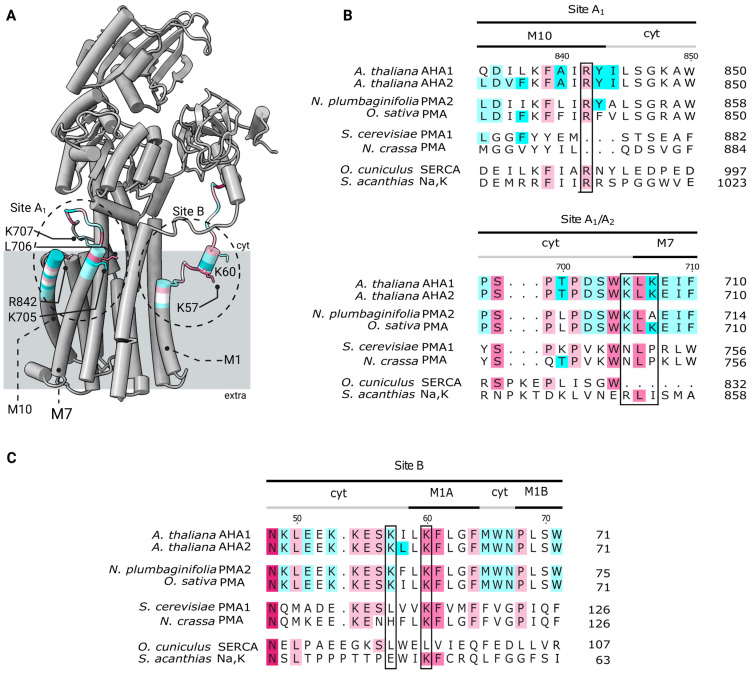
Location of predicted anionic phospholipid contact sites A_1_ and B in AHA2 and their sequence alignment in related P-type ATPases. (**A**) Cartoon representation of the AHA2. Patches of approx. 15 residues around the assigned contact residues are colored according to their sequence conservation based on AL2CO algorithm implemented into ChimeraX [32]. Preferentially interacting residues identified in the lipid–protein contact analysis are shown in atom representation. The approximate regions of the anionic lipid contacts sites, as identified in the lipid density maps, are marked with black, dashed circles. (**B**,**C**) Sequence alignment showing the conservation of residues based on AL2CO algorithm in selected P-type ATPase proton pumps from plants and yeasts as well as SERCA and Na,K-ATPase at the regions of the two selected anionic lipid contact sites identified in AHA2 (sites A1 and B). The sequences were obtained from the Uniprot database: *Arabidopsis thaliana* H^+^-ATPase isoform 1-2 (P20649 and P1945), shown in the first block, followed by the plant proton pumps *Nicotiana plumbaginifolia* H^+^-ATPase isoform 2 (Q42932), *Oryza sativa subsp. Japonica* H^+^-ATPase (Q7XPY2), and the two fungal pumps (Saccharomyces cerevisiae H^+^-ATPase isoform 1 (P05030), *Neurospora crassa* H^+^-ATPase (P07038). Finally, they are compared to the SERCA pump from rabbit *Oryctolagus cuniculus* (P04191) and the Na,K-ATPase from shark *Squalus acanthias* (Q4H132). Residues in black boxes are found to have enriched lipid contacts in the MD simulations.

## Data Availability

All the data are within the article and Appendix A. All the data are to be shared upon request.

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
