# Peer review of "Anionic Phospholipids Stimulate the Proton Pumping Activity of the Plant Plasma Membrane P-Type H+-ATPase"

_ijms, 2023, doi:10.3390/ijms241713106_

Round 1

Reviewer 1 Report

The authors presented a very useful fundamental work. I have just a few minor remarks.

1.       Line 50: “Phosphorylation of residues in the R-domain of AHA2…”. Please specify which domain we are talking about (i.e., the N-terminal or C-terminal one). It is also desirable to specify the residues that are phosphorylated.

2.       It is undesirable to use the term “transporter” in relation to ATPase (lines 53-54; also line 251: “For P-type ATPase cation transporters….”). It is better to use the term “pump”. (See the book “Pumps, Channels and Transporters: Methods of Functional Analysis”, R.J. Clarke and M.A.A. Khalid, eds., Wiley, 2015).

3.       Line 54. “ ….the isolated R-domain of AHA2 was shown to bind PS….” - Please specify which domain we are talking about (i.e., the N-terminal or C-terminal one).  

Reviewer 2 Report

The paper describes well executed research and is pleasure to read.

It is interesting to compare molecular motives of H+ ATPase from different plants and fungi and look at correlations to animal Ca2+ and Na+/K+ ATPases. What does this tell us about evolution of different life-forms?

Some small improvements:

 The acronyms should be explained the first time they appear in the text  (e.g. Line 55: “...bind PS in a protein…”. It might be helpful to include a list of acronyms and their meaning.

 The figures are good, except Fig. 3C, which is too small to resolve molecular detail. Perhaps this figure part could be shown as enlarged separate figure.
